# Amber Extract Suppressed Mast Cell-Mediated Allergic Inflammation via the Regulation of Allergic Mediators—An In Vitro Study

**Redoyan Refli** [1], **Neng Tanty Sofyana** [2], **Haruna Haeiwa** [3], **Reiko Takeda** [2,3], **Kazuma Okazaki** [3], **Marie Sekita** [3] and **Kazuichi Sakamoto** [1,2,*]

[1]  Graduate School of Science and Technology, University of Tsukuba, Tsukuba 305-8572, Japan
[2]  Faculty of Life and Environmental Sciences, University of Tsukuba, Tsukuba 305-8572, Japan
[3]  Kohaku Bio Technology Co., Ltd., Tsukuba 305-8572, Japan
*  Correspondence: sakamoto@biol.tsukuba.ac.jp; Tel.: +81-29-853-4676

**Abstract:** The various clinical approaches for treating allergy-related diseases have shown modest progress in low side effects and improved clinical outcomes. Therefore, finding alternative anti-allergic agents is crucial. The present study explored the anti-allergic effects of amber extract (fossilized tree resin) in RBL-2H3 mast cells stimulated with different allergens. In order to support the information on the inflammatory effect of the amber extract, NO production analysis on RAW 264.7 cells was conducted. β-Hexosaminidase release, an indicator of the efficacy of the amber extract in preventing mast cell activation and degranulation, reactive oxygen species (ROS) generation, and the effect of the amber extract on key cytokines production on RBL-2H3 cells, was evaluated. The results demonstrated that amber extract at concentrations up to 50 μg/mL had no cytotoxic effect on RAW 264.7 and RBL-2H3 cells. Amber extract inhibited NO production in RAW 264.7 cells. Treatment with amber extract significantly suppressed the release of β-hexosaminidase, especially at 50 μg/mL. Furthermore, amber extract suppressed the significantly increased ROS levels induced by allergen stimulation and allergy-associated cytokines. The results also suggested that amber extract exerts anti-allergic inflammatory effects by inhibiting the MAPK and NF-κB signaling pathways, resulting in decreased cytokines production. Thus, the amber extract is a promising anti-allergic agent.

**Keywords:** amber extract; anti-allergy; mast cells; ROS; degranulation; cytokines; MAPK; NF-κB

## 1. Introduction

Allergic diseases are inflammatory diseases that have become a global clinical health concern. About 10–20% of the world's population is affected by allergies. In recent decades, the rapid rise in the number of allergic patients has triggered extensive research, which has mainly focused on understanding the mechanisms of allergic immune responses and establishing new therapeutics for allergy-related diseases.

Mast cells play a key role in type I allergies and can cause anaphylaxis by degranulation [1]. The key mechanism of mast cell activation involves immunoglobulin (Ig)E-mediated allergic reactions through the FcεRI receptor [2]. Because inhibition of mast cell activation or degranulation is involved in the regulation of various IgE-mediated hypersensitivity reactions, mast cells have become an ideal target for finding anti-allergic drug candidates. Mast cell activation occurred through the cross-linking of FcεRI receptor-bound IgE. This activation triggers the initiation of the mitogen-activated protein kinase (MAPK) signaling pathway [3]. MAPK pathway activation includes the stimulation of proteins such as p38, extracellular signal-regulated kinase (ERK), and c-Jun N-terminal kinases (JNK). It also triggers mast cell degranulation, the secretion of β-hexosaminidase and histamine, and the increase in intracellular calcium [4]. Histamine and β-hexosaminidase are one of

the main mediators of allergic reactions because their releases are useful biomarkers of mast cell degranulation [5].

Calcium ionophores (e.g., A23187) are intracellular calcium compounds that can effectively activate the MAPK signaling pathway and mast cell degranulation [6]. The addition of phorbol myristate acetate (PMA) can increase intracellular calcium ions ($Ca^{2+}$) to enhance the effect of A23187 on mast cells. The increase in $Ca^{2+}$ in mast cells itself is also considered an important factor in mast cell degranulation, and it releases inflammatory cytokines.

Amber is a polymerized plant resin that was formed hundreds of millions of years ago in soil. It has primarily been used as jewelry, and during the Middle Ages, it was highly valued in Japan and used only by members of the imperial family. However, due to its bioactive properties, amber has also been used as a folk remedy in certain areas. Moreover, recent studies have demonstrated the potential of amber extracts to treat and prevent Alzheimer's disease [7], inflammation-related diseases [8], and obesity-related diseases [9]. Amber extracts also inhibit melanin production, promote collagen production [10], and have protective effects on neuronal cells against Parkinson's disease [11]. However, to our knowledge, there are limited reports on amber extract functionality regarding allergic diseases. Therefore, the present study aimed to define the anti-allergic inflammatory effects of amber extract and the molecular mechanism underlying such effects, including the inhibition of cytokines on a rat basophilic leukemia (RBL) cell line (RBL-2H3 cells). These cells were selected because they have been used to screen anti-allergic drug candidates in vitro.

## 2. Materials and Methods

### 2.1. Materials

The RBL-2H3 and macrophage (RAW 264.7) cells were procured from the American Type Culture Collection (ATCC; CRL-2256 and TIB-71, respectively). Dulbecco's modified Eagle's medium (DMEM), minimum essential medium (MEM), monoclonal anti-2,4-dinitrophenyl (DNP) IgE, calcium ionophore A23187, 3-(4,5-dimethylthiazol-2-yl)-2,5-diphenylterazolium bromide (MTT), Tyrode's salt, Griess reagent, and 4-nitrophenyl n-acetyl-β-d-glucosaminide (p-NAG) were purchased from Sigma-Aldrich (St. Louis, MO, USA). Fetal bovine serum (FBS), sodium pyruvate, and MEM non-essential amino acids (NEAA) were obtained from Gibco (Grand Island, NY, USA). The 2,4-dinitrophenyl hapten conjugated to bovine serum albumin (DNP-BSA) was obtained from Invitrogen (Carlsbad, CA, USA). PMA was supplied by Abcam (Cambridge, UK). Primary antibodies against cyclooxygenase (COX)-2 and β-actin were obtained from Cell Signaling Technology (Danvers, MA, USA), and interleukin (IL)-4 was obtained from Santa Cruz Biotechnology (Dallas, TX, USA).

### 2.2. Preparation of Amber Extract

Kaliningrad amber was excavated from mines located in Russia. Amber (50 g) was crushed into a powder and extracted in 50% ethanol at 40 °C. After stirring for 1 h, the extract was subjected to double filtration. The extract was then lyophilized to obtain a dried powder in a 3% yield provided by Kohaku Bio Technology, Tsukuba, Japan. The powder was dissolved in dimethyl sulfoxide and stored at −80 °C prior to use.

### 2.3. Cell Culture

RAW 264.7 cells were maintained in high-glucose DMEM supplemented with 10% FBS and RBL-2H3 cells in MEM supplemented with sodium pyruvate, NEAA, and 10% FBS. Both cell lines were incubated at 37 °C in a humidified atmosphere containing 5% $CO_2$.

### 2.4. Cell Viability Assay

RAW 264.7 and RBL-2H3 cells were seeded onto 96-well plates ($2.5 \times 10^4$ cells/mL) and treated with serial concentrations of the amber extract. The treated cells were incubated

in a 90% DMEM or MEM (RAW 264.7 and RBL-2H3 medium, respectively) plus 10% MTT solution for 4 h at 37 °C. After 4 h, a 10% sodium dodecyl sulfate (SDS) solution was added, and the cells were incubated at room temperature overnight. The optical density (OD) was measured at 570 nm using an Epoch2 microplate reader (BioTek, Winooski, VT, USA).

### 2.5. Determination of Nitric Oxide (NO) Production

RAW 264.7 cells were seeded and cultured on 24-well plates ($2.5 \times 10^5$ cells/mL). Lipopolysaccharide (LPS; 1 µg/mL), dexamethasone (DEX; 5 µg/mL), or amber extract (AE; 10, 25, and 50 µg/mL) were added to the cells. Next, 250 mL of $1\times$ Griess reagent was added, and the OD was measured at 540 nm.

### 2.6. Determination of β-Hexosaminidase

RBL-2H3 cells were seeded in 24-well plates ($2.5 \times 10^5$ cells/mL) and sensitized with 0.5 µg/mL anti-DNP IgE at 37 °C overnight. Next, the growth medium was replaced with modified Tyrode's assay buffer (Tyrode's salt, 20 mM HEPES, 1 mg/mL BSA, and 1 mg/mL NaHCO$_3$; pH 7.3), and the cells were then incubated. Cells were left untreated (control) or treated with quercetin (6 µg/mL) or amber extract (10, 25, and 50 µg/mL) for 2 h and then stimulated with 1 µg/mL DNP-BSA at 37 °C for 20 min. RBL-2H3 cells were stimulated with 30 ng/mL PMA and 350 ng/mL A23187 at 37 °C for 20 min, without IgE sensitization. The cell supernatant (50 µL) collected by centrifugation was transferred to a 96-well plate and incubated with 50 µL of 1 mM 4-nitrophenyl p-NAG in 0.1 M sodium citrate (pH 4.5) at 37 °C for 1 h. The reaction was terminated with 50 µL/well of glycine-NaOH 0.4 M (pH 10.7). Absorbance (OD) was measured at 405 nm using a microplate reader.

In order to measure the total amount of β-hexosaminidase release, the final cells were lysed using a buffer containing Triton X-100 solution (0.1% (*v/v*)) prior to incubation with the substrate. The activity of the supernatant was also measured using the same procedure. The percentage of β-hexosaminidase released was calculated by dividing the absorbance of the supernatant by the sum of the absorbances of the supernatant and cell lysate.

### 2.7. Intracellular ROS Accumulation

RBL-2H3 cells were seeded on a 6-well plate ($2 \times 10^5$ cells/mL) and sensitized with 0.5 µg/mL of IgE anti-DNP at 37 °C overnight. Cells were left untreated (control) or treated with quercetin (6 µg/mL) or amber extract (10, 25, and 50 µg/mL) and then incubated for 6 h. After incubation, the cells were stimulated with DNP-BSA at 37 °C for 1 h. RBL-2H3 cells were stimulated with PMA + A23187 at 37 °C for 4 h, without sensitization by IgE. Intracellular ROS concentrations were measured using 2′,7′ dichlorodihydrofluorescein diacetate (DCFDA) cellular ROS assay kits, as described by the manufacturer (Abcam, Cambridge, UK). Cells were stained with DCFDA for 20 min, and ROS production was measured on a fluorescence plate reader with intensities of Excitation (Ex)/Emission (Em): 485/535 nm. The representative image of stained cells was captured by a fluorescent microscope.

### 2.8. Western Blotting

RBL-2H3 cells were seeded and cultured in a 6-cm dish. The cells were then left untreated (control) or treated with quercetin (6 µg/mL) or amber extract (10, 25, and 50 µg/mL) for 6 h. After this period, cells were stimulated with PMA + A23187 and incubated for 8 h. The cells were then centrifuged, washed twice with cold phosphate-buffered saline, and lysed in RIPA buffer (150 mM NaCl, 1 mM EDTA, 50 mM Tris-HCl, 10 mM NaF, 1 mM Na$_3$VO$_4$, 1% Triton X-100, 0.1% SDS, 0.5% Na-deoxycholate, and protein inhibitor). Cells were lysed using an ultrasonic homogenizer and centrifuged to collect the supernatant. Protein quantification was conducted using a bicinchoninic acid protein assay kit (Wako Pure Chemical Industries, Ltd., Osaka, Japan). About 20 µg of protein was subjected to SDS-polyacrylamide gel electrophoresis analysis. After blocking for 1 h, the membrane was incubated with primary antibodies (IL-4, COX-2, and β-actin). The membrane was then incubated with the secondary antibodies horseradish peroxidase

(HRP)-conjugated secondary antibodies at room temperature for 1 h. LumiGLO® reagent (Cell Signaling Technology) was used to detect HRP. The protein bands were detected using an AE-9300 Ez-capture MG imaging system (ATTO Corporation, Tokyo, Japan).

*2.9. qRT-PCR*

Cells were seeded and cultured in a 6-cm dish. Cell groups were then left untreated (control) or treated with an allergen (DNP-BSA or PMA + A23187), quercetin, or amber extract (10, 25, and 50 μg/mL). Total RNA was isolated using RNAiso Plus (Takara Bio Inc., Kusatsu, Shiga, Japan). The total RNA was dissolved in diethyl pyrocarbonate-treated distilled water. A nanodrop was used to evaluate RNA purity by measuring absorbance at 260 and 280 nm. Complementary DNA was reverse-synthesized using the PrimeScript RT Reagent Kit with gDNA Eraser (Perfect Real Time, Takara Bio Inc.). The qRT-PCR was performed using the THUNDERBIRD® SYBR® qPCR Mix (Toyobo, Osaka, Japan). Amplification was performed under the following conditions: 95 °C for 30 s, followed by 45 cycles at 95 °C for 5 s and 60 °C for 30 s, and 95 °C for 15 s. Analysis was performed using the relative quantitative analysis software Multiplate RQ (Takara Bio Inc.). The relative gene expression levels were normalized to GAPDH expression levels. The primer sequences are listed in Table S1.

*2.10. Statistical Analysis*

Values are presented as the mean ± standard error of the mean (SEM). The statistical significance of all data was determined using Student's *t*-test. For RAW 264.7 cells, the data are expressed as mean ± SEM (+++ $p < 0.001$ vs. control, ** $p < 0.01$; *** $p < 0.001$ vs. LPS). For RBL-2H3 cells, the data are expressed as the mean ± SEM (+ $p < 0.05$, ++ $p < 0.01$, +++ $p < 0.001$ vs. control; * $p < 0.05$, ** $p < 0.01$, *** $p < 0.001$ vs. DNP-BSA or PMA+A23187 allergens).

## 3. Results

*3.1. Amber Extract at Concentrations up to 50 μg/mL Showed No Significant Cytotoxic Effects on RAW 264.7 and RBL-2H3 Cells*

The effect of the amber extract on cell viability was evaluated by MTT assay in RBL-2H3 and RAW 264.7 cells to ensure that its anti-allergic inflammatory effects were not due to cell death. Treatment with amber extract for 24 h at 0, 10, 12.5, 25, 50, and 75 μg/mL showed no significant cytotoxic effects but appeared to be slightly toxic at 100 μg/mL in RAW 264.7 cells. As for RBL-2H3 cells, no toxicity effects were detected at the concentration of 10, 12.5, 25, and 50 μg/mL. However, amber extract at concentrations of 75 and 100 μg/mL in RBL-2H3 cells was slightly toxic. In addition, amber extract at concentrations up to 50 μg/mL led to the increased growth of RAW 264.7 cells (Figure 1a,b). Based on these results, concentrations ranging from 10 to 50 μg/mL were selected for further studies.

**(a)** 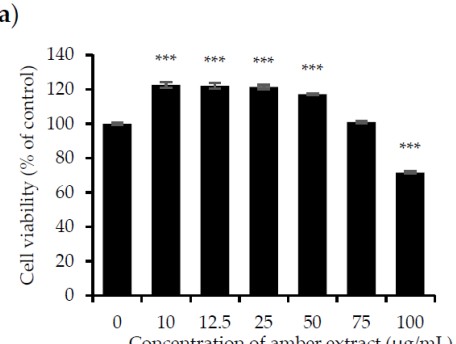 **(b)** 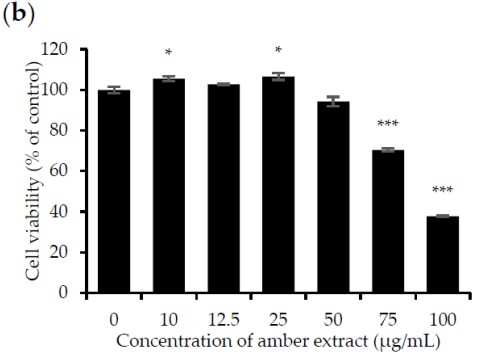

**Figure 1.** Effect of amber extract on cell viability: (**a**) Cell viability of RAW 264.7 cells. (**b**). Cell viability of RBL-2H3 cells. After culturing the cells with amber extract (0, 10, 12.5, 25, 50, 75, and 100 μg/mL) for 24 h, cell viability was measured using the MTT assay (* $p < 0.05$, *** $p < 0.001$ vs. control).

### 3.2. Amber Extract Significantly Inhibited NO Production

NO is considered an important factor in the inflammatory response and defense against infectious organisms. In order to investigate the potential anti-inflammatory effects of the amber extract on LPS-stimulated RAW 264.7 cells, we explored the inhibitory effect of the amber extract on NO production using a Griess reagent. The amber extract significantly suppressed the generation of NO in a concentration-dependent manner (Figure 2), especially at 50 μg/mL. This result suggested a potential anti-inflammatory effect of amber extract via inhibiting NO production.

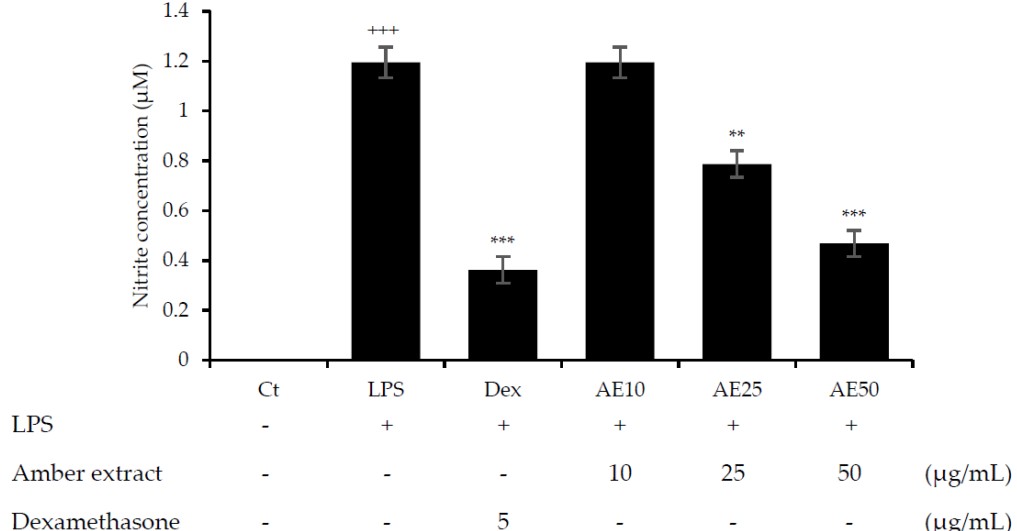

**Figure 2.** Effect of amber extract on NO production. After culturing the cells with LPS, dexamethasone (DEX), or amber extract (10, 25, and 50 μg/mL) for 24 h, NO production was measured using the Griess reagent. Data are expressed as mean ± standard error of the mean (SEM) (+++ $p < 0.001$ vs. control; ** $p < 0.01$, *** $p < 0.001$ vs. LPS).

### 3.3. Effect of Amber Extract on β-Hexosaminidase Secretion in RBL-2H3 Cells

β-hexosaminidase is present in the cytoplasmic granules of mast cells, and its activation causes degranulation. β-hexosaminidase is secreted in response to allergic reactions, and its release can be used to quantify the extent of degranulation [12]. Pre-treatment with amber extract significantly suppressed degranulation in either 1 μg/mL DNP-BSA- or 30 ng/mL PMA plus 350 ng/mL A23187 (PMA + A23187)-stimulated RBL-2H3 cells (Figure 3a,b). In addition, the amber extract reduced the degranulation in RBL-2H3 cells in a dose-dependent manner, and at 50 μg/mL, PMA + A23187-stimulated RBL-2H3 cells showed more effective inhibition of β-hexosaminidase release than the positive control quercetin. The DNP-BSA-stimulated RBL-2H3 cells showed similar inhibition of β-hexosaminidase release as the positive control quercetin at 50 μg/mL. These results suggested that the amber extract could exert anti-allergic effects on RBL-2H3 cells.

### 3.4. Amber Extract Inhibited Reactive Oxygen Species (ROS) Generation in RBL-2H3 Cells

The effects of amber extract (10, 25, and 50 μg/mL) in either DNP-BSA- or PMA+A23187-stimulated RBL-2H3 cells were investigated (Figure 4a–d). The amber extract decreased the significantly increased ROS generation in a dose-dependent manner, especially at 50 μg/mL, in both allergen stimulations. This inhibition was similar to that of the positive control, quercetin. These results indicated the promising effect of the amber extract on promoting cellular antioxidant capacity.

### 3.5. Amber Extract Inhibited the Inflammatory Mediators

Inflammation mediators, such as IL-4 and COX-2, are involved in the late-phase reaction of allergy. Western blot analysis was used to investigate the effect of the amber

extract on the activation of IL-4 and COX-2. RBL-2H3 cells stimulated with PMA + A23187 showed decreased levels of both COX-2 and IL-4 in a dose-dependent manner (Figure 5). This suggested a potent inhibitory effect of the amber extract on the levels of inflammatory mediators (COX-2 and IL-4).

### 3.6. Effects of Amber Extract on Cytokine-, MAPK-, and Nuclear Factor (NF)-κB-Related Gene Expressions in RBL-2H3 Cells Stimulated with PMA + A23187

The expression levels of the genes encoding IL-4, IL-13, tumor necrosis factor (TNF)-α, and COX-2 were measured by quantitative real-time PCR (qRT-PCR) (Figure 6A–D). PMA + A23187 stimulation significantly increased the expression of IL-4 (Figure 6A), which was significantly decreased by treatment with amber extract at 50 μg/mL. Furthermore, amber extract at 50 μg/mL showed almost similar inhibition as the positive control quercetin. The amber extract decreased the expression of IL-13 (Figure 6B) in PMA + A23187-stimulated RBL-2H3 cells. The inhibition was especially remarkable at 25 and 50 μg/mL, which showed similar inhibition as quercetin. The results also demonstrated that the amber extract significantly suppressed the expression of TNF-α compared to the PMA + A23187-stimulated cells (Figure 6C). The expression of COX-2 tended to decrease in a dose-dependent manner in the amber extract-treated cells compared to the PMA + A23187-stimulated cells (Figure 6D).

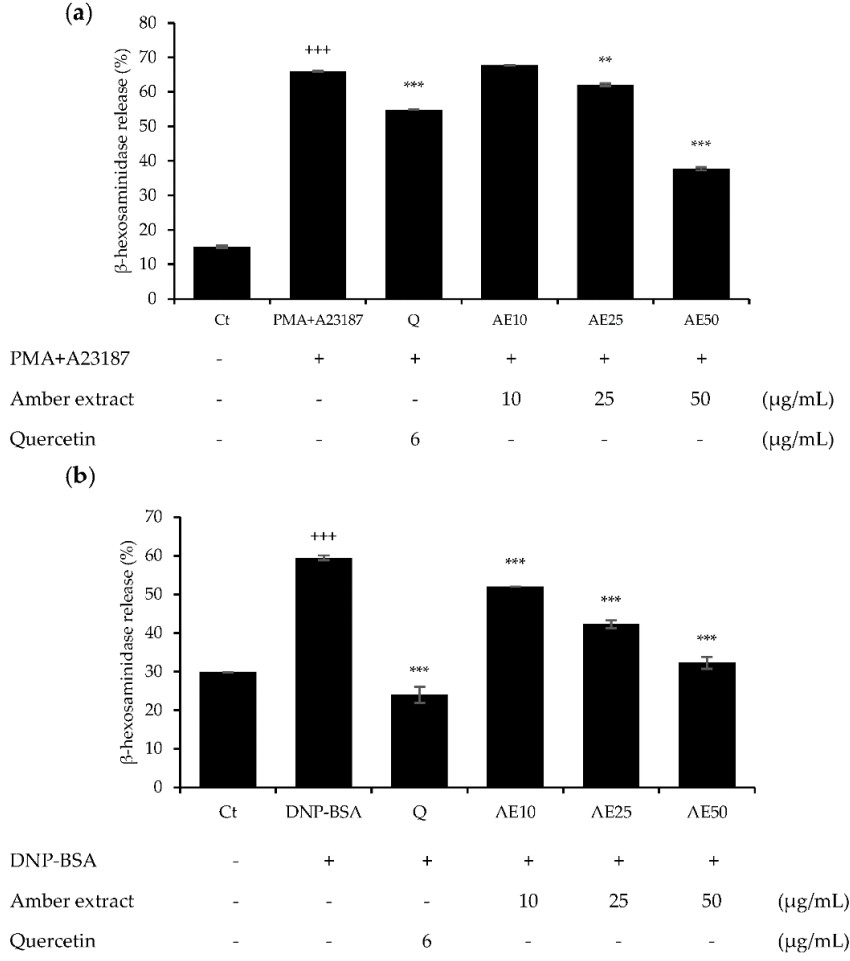

**Figure 3.** Effects of amber extract on β-hexosaminidase secretion in RBL-2H3 cells: (**a**) β-hexosaminidase secretion in PMA+A23187-stimulated RBL-2H3 cells. (**b**) β-hexosaminidase secretion in DNP-BSA-stimulated RBL-2H3 cells. β-hexosaminidase release was measured after incubation in assay buffer and stimulation with DNP-BSA or PMA+A23187. Data are expressed as mean ± SEM (+++ $p < 0.001$ vs. control; ** $p < 0.01$, *** $p < 0.001$ vs. DNP-BSA or PMA+A23187).

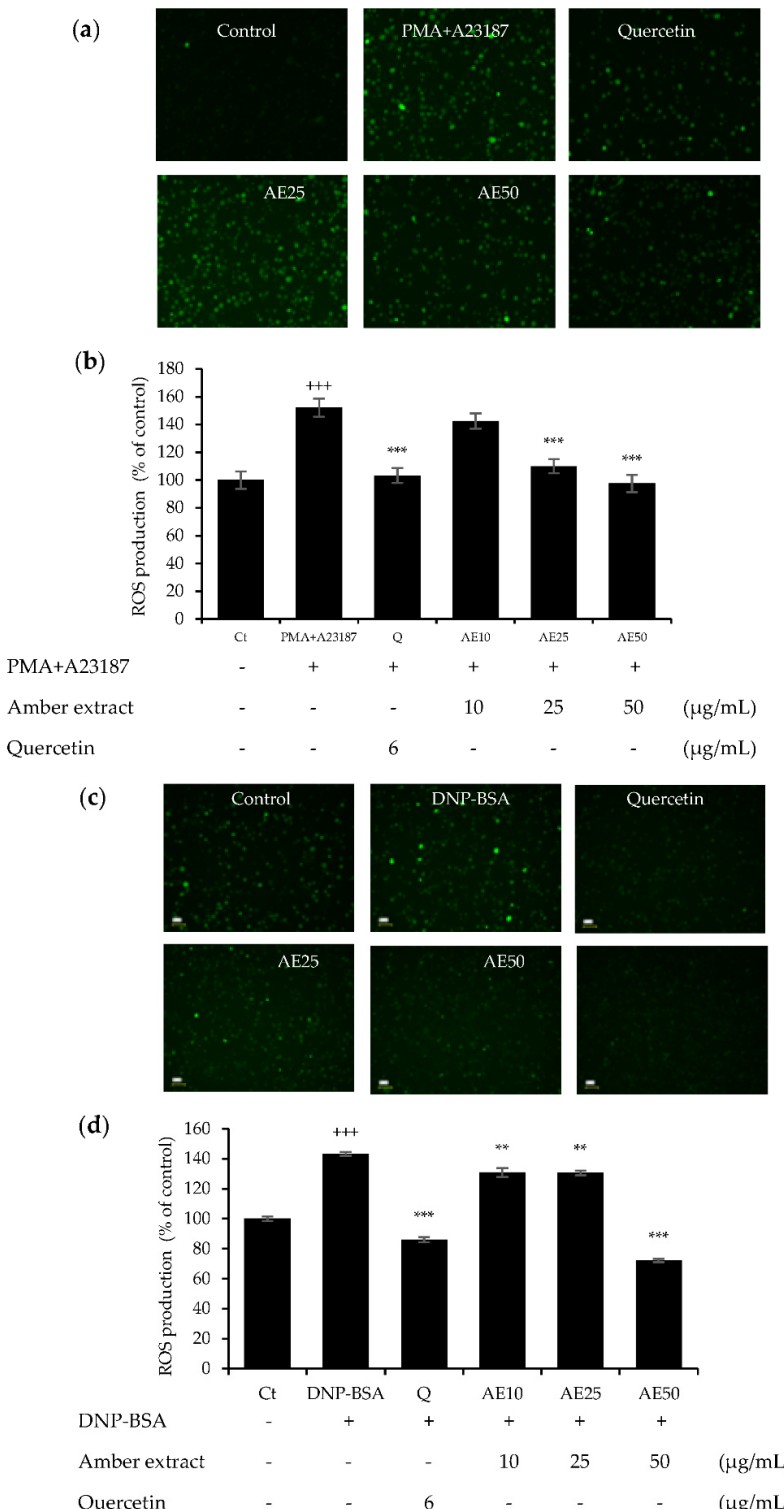

**Figure 4.** Effect of amber extract on ROS generation: (**a**) ROS generation staining in PMA+A23187-stimulated RBL-2H3 cells. (**b**) ROS production in PMA+A23187-stimulated RBL-2H3 cells. (**c**) ROS generation staining in DNP-BSA-stimulated RBL-2H3 cells. (**d**) ROS production in DNP-BSA-stimulated RBL-2H3 cells. Cells were stained with 2′,7′ dichlorodihydrofluorescein diacetate (DCFDA) for 20 min. The cells were photographed at 20× magnification using a fluorescent microscope. The presence of ROS is indicated in green. ROS production was measured on a fluorescence plate reader at Ex/Em = 485/535 nm. Data are expressed as mean ± SEM (+++ $p < 0.001$ vs. control; ** $p < 0.01$, *** $p < 0.001$ vs. DNP-BSA or PMA+A23187).

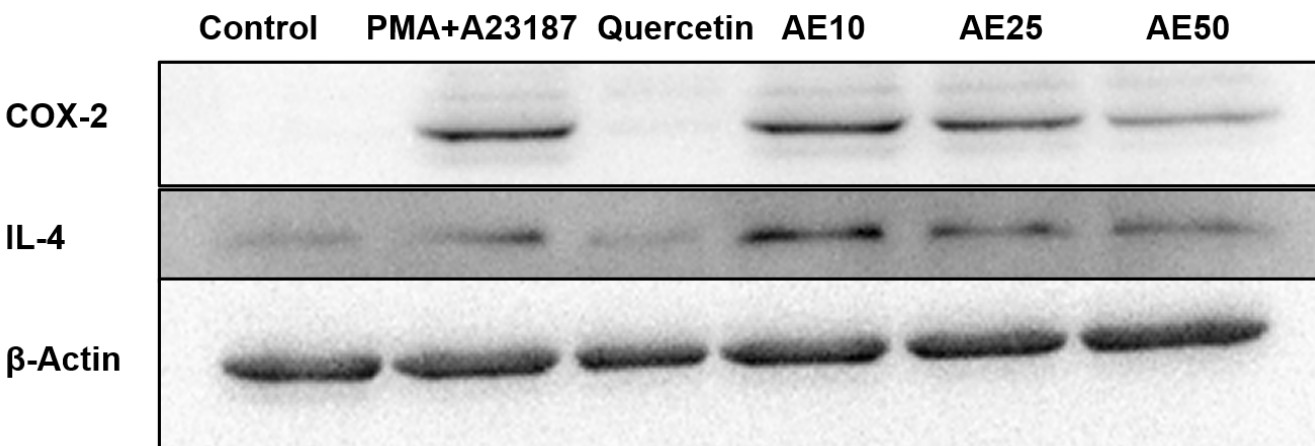

**Figure 5.** Effect of amber extract on the activation of cytokine-related proteins. After culturing the cells (untreated control), allergen (PMA + A23187), quercetin (6 μg/mL), or amber extract (10, 25, and 50 μg/ mL) were added, the levels of IL-4 and COX-2 production were measured by Western blotting.

The amber extract significantly suppressed the expression of ERK1 and tended to decrease the mRNA expression level of key MAPK factor genes (ERK2, JNK, and p38) (Figure 6E). The expression of IκB-α tended to increase with the treatment of amber extract. In addition, the amber extract significantly suppressed the expression of NF-κB and tended to suppress the expression of NF-κB p65 (Figure 6F).

### 3.7. Effects of Amber Extract on Cytokine- and NF-κB-Related Gene Expressions in RBL-2H3 Cells Stimulated with DNP-BSA

The RBL-2H3 cells were previously stimulated by DNP-BSA. The results demonstrated that the amber extract suppressed the expression of IL-4 compared to the DNP-BSA-stimulated group in a dose-dependent manner, especially at 50 μg/mL (Figure 7A). The amber extract significantly suppressed FcεRI-α expression on the mast cell surface (Figure 7B). The expression of IL-6 (Figure 7C) decreased in a dose-dependent manner following treatment with the amber extract. The inhibition was especially remarkable and was similar to that of quercetin at 50 μg/mL. The amber extract suppressed the expression of IL-13, TNF-α, and COX-2 (Figure 7D–F) in RBL-2H3 cells stimulated with DNP-BSA. The expression of IκB-α significantly increased with the treatment of amber extract. In addition, the amber extract significantly inhibited the increased expression of NF-κB and NF-κB p65 (Figure 7G).

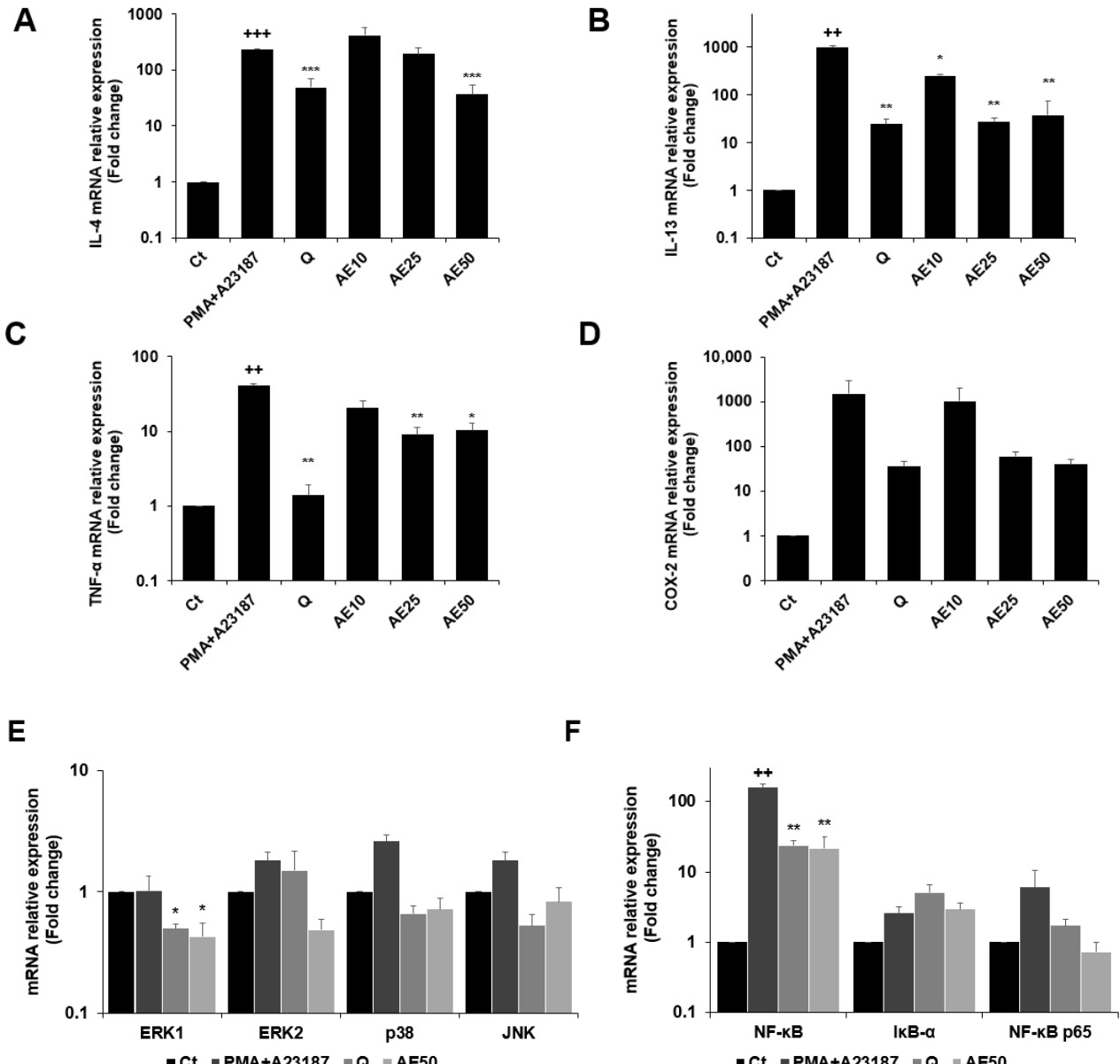

**Figure 6.** Effects of the amber extract on cytokine-, MAPK-, and NF-κB-related gene expressions of RBL-2H3 cells stimulated by PMA + A23187 for 4 h. (**A**) The mRNA expression levels of IL-4. (**B**) The mRNA expression levels of IL-13. (**C**) The mRNA expression levels of TNF-α. (**D**) The mRNA expression levels of COX-2. (**E**) The mRNA expression levels of ERK1/2, JNK, and p38. (**F**) The mRNA expression levels of NF-κB, IκB-α, and NF-κB p65 were measured by qRT-PCR. Data are expressed as mean ± SEM (++ $p < 0.01$, +++ $p < 0.001$ vs. control; * $p < 0.05$, ** $p < 0.01$, *** $p < 0.001$ vs. PMA + A23187).

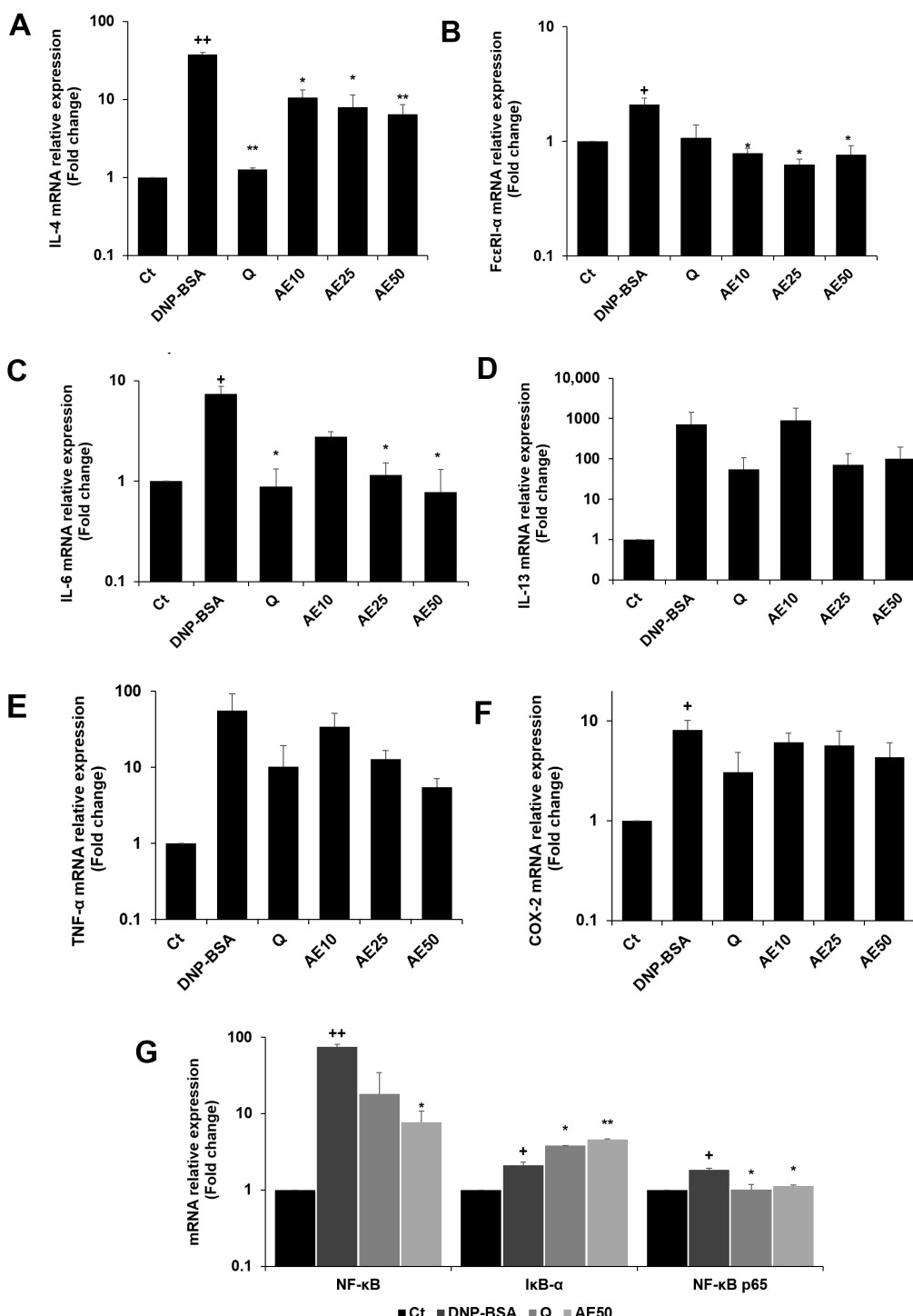

**Figure 7.** Effects of the amber extract on cytokine- and NF-κB-related gene expressions of RBL-2H3 cells stimulated by DNP-BSA for 1 h. (**A**) The mRNA expression levels of IL-4. (**B**) The mRNA expression levels of FcεRI-α. (**C**) The mRNA expression levels of IL-6. (**D**) The mRNA expression levels of IL-13. (**E**) The mRNA expression levels of TNF-α. (**F**) The mRNA expression levels of COX-2. (**G**) The mRNA expression levels of NF-κB. IκB-α, and NF-κB p65 were measured by qRT-PCR. Data are expressed as mean ± SEM (+ $p < 0.05$, ++ $p < 0.01$ vs. control, * $p < 0.05$, ** $p < 0.01$ vs. DNP-BSA).

## 4. Discussion

Amber is a fossil resin that originated from plants. Their dominant compound of amber is inherited from the resins characterized by polymerization processes occurring during burial. The polymer of amber is divided into polymers of labdatriene carboxylic acids (main

compound: succinite), polymers of cadinene (main compound: polycadinene), polymers of styrene (main compound: styrene and cinnamic acid), and non-polymeric ambers (main compound: sesquiterpenoid) [13]. Succinic acid has been reported to inhibit systemic anaphylaxis in mast cells and exhibit therapeutic effects on gestational hypertension [14,15]. Cinnamic acid has antimicrobial and antioxidant activity [16]. Moreover, in the previous study, sesquiterpenoids were reported to have anti-inflammatory effects due to the β-eudesmol structure and lactone ring-bearing sesquiterpene [17]. In addition, one of the insoluble polymeric compounds of amber is abietic acid, which is the most characteristic component of amber. This compound confers mechanical properties of amber that do not vary from one sample of amber to another [18]. One functional compound related to abietic acid, dehydroabietic acid, has been reported to suppress chronic inflammation in obese mice [19]. However, compounds that exhibit anti-allergic effects from amber have not been fully understood.

Inflammatory disorders are allergic responses to allergens. Macrophages activated by LPS are key immune cells involved in the initiation of inflammation. Therefore, LPS-induced macrophages are commonly used to assess the anti-inflammatory effects of various agents. In the present study, RAW 264.7 macrophages were used to observe the anti-inflammatory activity of the amber extract. NO is an endogenous free radical present at low levels in the human body. Suppression of NO production is usually used as an important treatment for inflammatory diseases [20]. Treatment with the amber extract inhibited LPS-induced overproduction of NO in RAW 264.7 macrophage cells. This result indicates that the amber extract exerts anti-inflammatory effects by inhibiting NO production.

Mast cell degranulation is a key mediator in the early phase of allergic inflammation. This process is mediated by both the immune and non-immune pathways. Activation of the immune pathway is related to the aggregation of specific surface receptors, such as FcεRI, binding to several complexes of antigens with IgE [21]. The non-immune pathways of mast cells are activated polycationic mast cell activators, neuropeptides, cytokines and chemokines, anaphylatoxins, calcium ionophores (A23187 or ionomycin), and other substances [22]. β-Hexosaminidase is a granular protein that plays a central role in the progression of allergic reactions [23]. The release of β-hexosaminidase is used as a marker of the extent of degranulation [24]. Therefore, in the present study, we investigated whether amber extract could inhibit the degranulation of RBL-2H3 mast cells by measuring the production of β-hexosaminidase after stimulation with DNP-BSA or PMA + A23187, which are representatives of immune and non-immune pathways, respectively. As the release of β-hexosaminidase was significantly increased upon stimulation with both DNP-BSA and PMA + A23187 in RBL-2H3 cells, the amber extract was confirmed to significantly inhibit DNP-BSA- or PMA+A23187-stimulated β-hexosaminidase in a dose-dependent manner, particularly at 50 µg/mL. These results suggest that amber extract exerts anti-allergic effects via both immune and non-immune pathways.

Mast cells are exposed to an oxidative environment during allergic and inflammatory reactions. ROS can stimulate the production of some pro-inflammatory cytokines and participate in the regulation of innate immunity. ROS plays an important role in mast cell degranulation in FcεRI signaling [25]. To counteract the deleterious effects of ROS generation, cells have developed elaborate antioxidant defense mechanisms [26]. As shown in Figure 4a–d, amber extract suppressed the generation of ROS in RBL-2H3 cells upon stimulation with DNP-BSA or PMA + A23187. These results indicate that amber extract reduces ROS production and plays a key role in susceptibility to allergic sensitization. The antioxidant effect was exerted by the metabolites in the extract. Poulin and Helwig provided direct molecular evidence of the structural role of amber's succinic acid [27]. This compound has a favorable impact on intracellular medium oxygenation, preventing excessive lipid peroxidation and suppressing antioxidant systems [28]. Furthermore, an increase in oxidative stress has also been shown to be involved in the activation of Lyn, Syk, and MAPK signaling in allergic reactions. Therefore, the amber extract may exert an anti-allergic effect via cellular antioxidants in the presence of succinic acid.

Inflammatory mediators such as IL-4 and COX-2 are involved in allergic reactions. IL-4 is essential for IgE [29]. In allergic inflammatory diseases, IL-4 is considered a key indicator for reducing allergy symptoms [30]. IL-4 modulates the inflammatory response by affecting cytokine production, which is an important therapeutic step [31]. COX-2 is primarily expressed during the inflammatory response [32]. The present study demonstrated the dose-dependent effects of the amber extract on PMA + A23187-stimulated IL-4 and COX-2 production. These results indicated that the amber extract exerts anti-allergic effects by controlling cytokine levels in RBL-2H3 cells.

The MAPK and NF-$\kappa$B pathways are the two major mechanisms that regulate the production of inflammatory cytokines. MAPK signaling is typically regarded as a mediator of late-phase reactions. Activated mast cells trigger the production of cytokines from the induction of cytokine gene transcription in these cells [33]. Activation of the MAPK signaling pathway affects transcription factor activity in the secretion of cytokines, including IL-4 and IL-13. This activation also increases intracellular $Ca^{2+}$, which triggers mast cell degranulation [6]. Therefore, numerous allergic studies have focused on compounds that downregulate the MAPK pathway [31]. In the present study, the amber extracts profoundly affected DNP-BSA- and PMA+A23187-induced allergic reactions by significantly downregulating the expression of IL-4. IL-4 is involved in the late phase of allergy by modulating the inflammatory response, which affects adhesion molecule expression and cytokine production in endothelial cells [34,35]. These results support the IL-4 protein production, which also exhibited dose-dependent effects of the amber extract against both DNP-BSA- and PMA + A23187-stimulated IL-4 production. These results indicate that the amber extract may be a promising new anti-allergy inflammatory agent.

In the present study, IL-13 expression in RBL-2H3 cells stimulated by PMA + A23187 was significantly downregulated by treatment with the amber extract. In addition, amber extract tended to inhibit the IL-13 expression associated with the downregulation of Fc$\varepsilon$RI-$\alpha$ expression in RBL-2H3 cells stimulated by DNP-BSA. IL-13 is a pleiotropic cytokine that stimulates mast cell proliferation and upregulates Fc$\varepsilon$RI on the mast cell surface [36,37]. Thus, further identification and regulation of key targets in the IgE-Fc$\varepsilon$RI signaling pathways might be effective approaches for the treatment of allergic diseases.

When antigens trigger IgE-sensitized mast cells, Fc$\varepsilon$RI aggregation activates the phosphorylation of Lyn kinase, leading to the binding of Syc kinase (both belonging to the tyrosine kinase family). Syk is considered an important molecule in the allergic response to upstream Fc$\varepsilon$RI signaling [38,39]. The activation of Syk also causes the activation of downstream (MAPK-dependent) signaling pathways, which play a crucial role in the expression of allergic and pro-inflammatory mediators, including IL-6, TNF-$\alpha$, and COX-2 [40]. IL-6 is a primary inducer of acute-phase reaction proteins in chronic inflammatory diseases [41]. TNF-$\alpha$ is also a potent pro-inflammatory mediator secreted upon antigen stimulation in IgE-sensitized mast cells [42]. The present study demonstrated that the amber extract downregulated the expression level of IL-6 and suppressed the increase in TNF-$\alpha$ and COX-2 in DNP-BSA-stimulated RBL-2H3 cells. In addition, the amber extract significantly inhibited the upregulation of TNF-$\alpha$ and COX-2 in PMA + A23187-stimulated RBL-2H3 cells. These results imply that amber extract exerts anti-allergic effects by controlling cytokine gene expression levels in RBL-2H3 cells.

MAPK signaling pathway activation includes the stimulation of ERK1/2, JNK, and p38 and increases intracellular $Ca^{2+}$, triggering mast cell degranulation as well as the release of histamine and $\beta$-hexosaminidase [43]. ERK1/2 is activated by various stimuli and phosphorylates various transcription factors. JNK and p38 are activated by inflammatory cytokines during stress responses due to ROS generation [44]. JNK is also involved in transcription [45]. Calcium ionophores (e.g., A23187) are effective activators of the MAPK signaling pathway and mast cell degranulation [5]. Therefore, in the present study, we investigated the mRNA expression levels of ERK1/2, JNK, and p38 to explore the effect of the amber extract on the allergy mechanism, especially in the MAPK signaling pathway of RBL-2H3 cells stimulated by PMA + A23187. The amber extract significantly suppressed

ERK1 and tended to downregulate ERK2, JNK, and p38 expression levels after PMA + A23187 stimulation of RBL-2H3 cells. The molecular mechanisms and targets of the amber extract in the regulation of allergic responses remain to be identified. However, as the amber extract decreased the activation level of key MAPK signaling targets, its beneficial anti-allergy properties may be related to the downregulation of cytokine mediators.

MAPK activation leads to the activation of downstream NF-κB signaling. The present study showed that the amber extract significantly suppressed the expression level of NF-κB and tended to suppress NF-κB p65 in RBL-2H3 cells stimulated by PMA + A23187. NF-κB activation may also be inhibited by the upregulation of IκB-α expression levels. Furthermore, in RBL-2H3 cells stimulated by DNP-BSA, the amber extract significantly suppressed NF-κB, NF-κB p65. The reduction in NF-κB activation may therefore be due to the increase in IκB-α expression levels and inhibition of MAPK activation. Inhibition of the NF-κB pathway can further inhibit the release of cytokines. This mechanism is summarized in Figure 8.

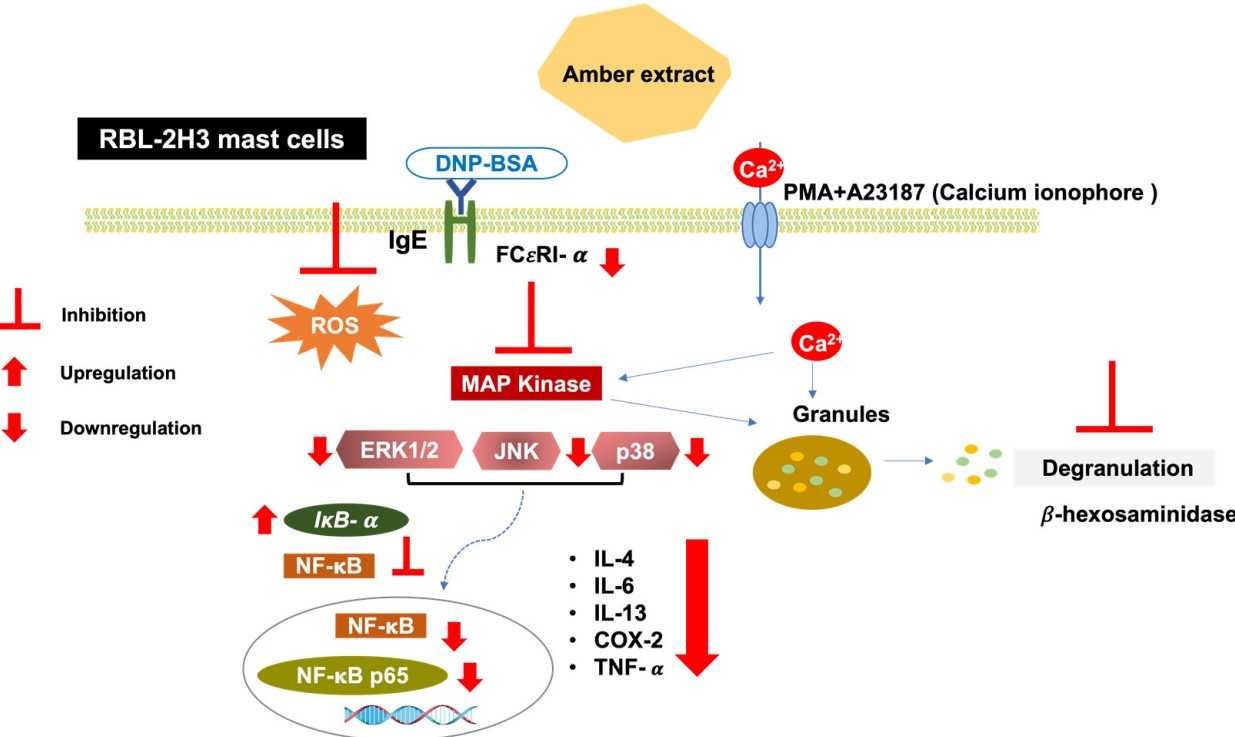

**Figure 8.** Schematic pathway summarizing the anti-allergy inflammatory effect of amber extract on RBL-2H3 cells. Antigen/IgE cross-linking with the mast cells' specific receptor FCεRI and calcium ionophore (A23187) induction activates the MAPK pathway. Amber extract decreases MAPK and NF-κB pathway activation and therefore diminishes mast cell degranulation and cytokines production.

Dehydroabietic acid is one of the main constituents of amber resin, which has been reported to suppress the activity of Syk and Src in NF-κB inflammatory signaling pathways [46]. Moreover, monoterpenoids, which are also compound found in the Baltic amber, possess a cinnamoyl moiety attached to the monoterpene unit (borneol), which is likely the active part of their structure that mediates their anti-allergy activity [17]. Thus, the anti-allergic activity of amber extract from this study may be due to the presence of the components mentioned above.

Our results provide valuable information that amber extract may be useful for the prevention and treatment of allergic diseases. However, more research regarding its bioactive composition is needed to elucidate the mechanism and better understand which bioactive compounds are responsible for the anti-inflammatory effects of amber.

## 5. Conclusions

In conclusion, the present study demonstrates that amber extract inhibits NO production in RAW 264.7 cells. ROS generation, antigen-induced degranulation, protein secretion of IL-4 and COX-2, mRNA expression of cytokine genes, MAPK, and NF-κB signaling representative genes in RBL-2H3 cells stimulated by DNP-BSA and PMA + A23187 were also demonstrated. These results indicate that amber extract has potential anti-inflammatory effects as a promising therapeutic agent for allergy-related diseases. However, further studies are required to optimize its utility in the pharmacological industry.

**Supplementary Materials:** The following supporting information can be downloaded at: https://www.mdpi.com/article/10.3390/nutraceuticals3010006/s1, Table S1: Primers list for the qRT-PCR.

**Author Contributions:** Conceptualization, K.S.; Methodology, K.S. and R.R.; Sample preparation, R.T., K.O., M.S. and H.H.; Analysis, R.R.; Data curation, R.R., N.T.S. and K.S.; Writing—original draft preparation, R.R.; Writing—review and editing, R.R., N.T.S. and K.S.; Supervision, K.S.; Project administration, K.S.; Funding acquisition, K.S. All authors have read and agreed to the published version of the manuscript.

**Funding:** This research was supported in part by Grants-in-Aid for Scientific Research and Education from the University of Tsukuba, Japan; Kohaku Bio Technology Co., Ltd., Tsukuba, Japan; and the Japan Science and Technology, SPRING (Grant no. JPMJSP2124). The sponsors had no role in the design, execution, interpretation, or writing of this manuscript.

**Institutional Review Board Statement:** Not applicable.

**Informed Consent Statement:** Not applicable.

**Data Availability Statement:** Data are contained within the article or Supplementary Materials.

**Acknowledgments:** The authors would like to thank Kohaku Bio Technology Co., Ltd., Tsukuba, Japan, for providing amber samples.

**Conflicts of Interest:** The authors declare no conflict of interest.

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
