# Peer review of "Amber Extract Suppressed Mast Cell-Mediated Allergic Inflammation via the Regulation of Allergic Mediators—An In Vitro Study"

_nutraceuticals, doi:10.3390/nutraceuticals3010006_

Round 1

Reviewer 1 Report

Interesting manuscript, but I have some concerns that need to be corrected through revisions, before further consideration.

1. The title should be "Amber extract suppressed mast cell-mediated allergic inflammation via the regulation of allergic mediators - An In VItro Study"

2. Introduction there must be a discussion regarding amber as a food, even though this is related to the author's hope that amber is used as a functional food.

In the introduction it should also be mentioned, what are the bioactive ingredients of this Amber extract.

3. Methods must be detailed, check thoroughly and provide detail by detail,

For example, how are samples obtained and then authenticated and identified by the herbarium? provide coordinates maps from the sampling location (can be seen on google maps).

4. In the graphical abstract and figure 8, you should include the original image of the amber extract or the amber itself.

Author Response

Reviewer: 1

Comments and Suggestions for Authors

Interesting manuscript, but I have some concerns that need to be corrected through revisions, before further consideration.

Comment 1: The title should be "Amber extract suppressed mast cell-mediated allergic inflammation via the regulation of allergic mediators - An In VItro Study"

Response: Thank you very much for your advice. We accordingly revised the title with "Amber extract suppressed mast cell-mediated allergic inflammation via the regulation of allergic mediators - an in vitro study".

Comment 2: Introduction there must be a discussion regarding amber as a food, even though this is related to the author's hope that amber is used as a functional food. In the introduction it should also be mentioned, what are the bioactive ingredients of this Amber extract.

Response: Thank you very much for pointing this out. We are thankful to the reviewer for the valuable suggestions given. Currently, we are working on identifying more of the unknown components of amber extract because the composition has not been elucidated yet. The purification and composition analysis of amber is carried out by the other group in this project. So, instead of introduction, we have already mentioned about the information of the common bioactive ingredients of amber in the discussion part, page 11 line 334: “The polymer of amber divided into polymers of labdatriene carboxylic acids (main compound; succinite), polymers of cadinene (main compound; polycadinene), polymers of styrene (main compound; styrene and cinnamic acid), and non-polymeric ambers (main compound; sesquiterpenoid)”.

We have also mentioned about the expected compound responsible in the allergic effects of amber extract onpage 12 line 387: “Therefore, amber extract may exert an anti-allergic effect via cellular antioxidants in the presence of succinic acid”. We have also mentioned that bioactive components, DAA (Dehydroabietic acid) and monoterpenoid might be responsible for allergic activity of amber extract in the page 13 line 464:” and line 466: “Dehydroabietic acid is one of the main constituents of amber resin which has been reported to suppress the activity of Syk and Src in NF-κB inflammatory signaling pathways [48]. Moreover, monoterpenoid, which also compound found in the Baltic amber, possess a cinnamoyl moiety attached to the monoterpene unit (borneol), which is likely the active part of their structure that mediates their anti-allergy activity [17]. Thus, the anti-allergic activity of amber extract from this study may be due to the present of the components mentioned above”.

Comment 3: Methods must be detailed, check thoroughly and provide detail by detail. For example, how are samples obtained and then authenticated and identified by the herbarium? provide coordinates maps from the sampling location (can be seen on google maps).

Response: Thank you very much for your suggestion. We have checked the methods and added some details information in the method for intracellular ROS accumulation procedure. We added the sentence “Cells were stain with DCFDA for 20 min and ROS production was measured on a fluorescence plate reader with intensities of Excitation (Ex)/Emission (Em): 485/535 nm. The representatives’ image of stained cells was captured by fluorescent microscope” in page 3 line 155.

The amber used in this experiment is custom-made products from Kaliningrad, Russia, imported from Esprima Company (Osaka, Japan). Therefore, we are sorry for not being able to provide the explanation about the sampling and identification method.

Comment 4: In the graphical abstract and figure 8, you should include the original image of the amber extract or the amber itself.

Response: Thank you very much for your comments and suggestions. We are sorry because the original image of amber is not provided this time because we used the custom-made products of amber extract from our collaborator Kohaku Bio Technology Co. Ltd.

Reviewer 2 Report

L. 178 

you should mention the other concentrations 75 & 100 μg/mL  

Fig. 4 C L. 226

the stain in AE25 is not matched with the graph result

Author Response

Reviewer: 2

Comment 1: L. 178. You should mention the other concentrations 75 & 100 μg/mL  

Response: Thank you very much for your suggestion. We have revised the sentence from

Treatment with amber extract for 24 h at 0, 10, 12.5, 25, and 50 μg/mL showed no significant cytotoxic effects but appeared to be slightly toxic at 100 μg/mL in RAW 264.7 cells and 75 and 100 μg/mL in RBL-2H3 cells.” to“Treatment with amber extract for 24 h at 0, 10, 12.5, 25, 50, and 75 μg/mL showed no significant cytotoxic effects but appeared to be slightly toxic at 100 μg/mL in RAW 264.7 cells. As for RBL-2H3 cells, no toxicity effects were detected at the concentration of 10, 12.5, 25, and 50 μg/mL. However, amber extract at the concentration of 75 and 100 μg/mL in RBL-2H3 cells were slightly toxic”. Please refer to page 4 line 209-213.

Comment 2: Fig. 4 C L. 226, the stain in AE25 is not matched with the graph result

Response: Thank you for raising this important point. The graph result on Fig. 4D is not the interpretation of the images staining. The graph was obtained from the measurement of ROS production on a fluorescence microplate reader. After reading the fluorescence, we captured the representative stained cells image using fluorescence microscope.  Therefore, we can state that the explanation of the results in the manuscript are match based on each figure. I am sorry for making a confusion about the data. To make it clear, I added more detail explanation in the methodCells were stain with DCFDA for 20 min and ROS production was measured on a fluorescence plate reader with intensities of Excitation (Ex)/Emission (Em): 485/535 nm. The representatives’ image of stained cells was captured by fluorescent microscope” please refer to page 3 line 155.

Reviewer 3 Report

The approach is well designed and it was so appropriate to utilize 2 cell lines for testing the amber extract activity. The data and results are properly presented and eligibly discussed. They concluded that further studies are required to elucidate the chemical entities of the amber extract and correlate it to the biological activities. 

I recommend accepting the manuscript.

Author Response

Reviewer: 3

Comments and Suggestions for Authors

The approach is well designed and it was so appropriate to utilize 2 cell lines for testing the amber extract activity. The data and results are properly presented and eligibly discussed. They concluded that further studies are required to elucidate the chemical entities of the amber extract and correlate it to the biological activities. I recommend accepting the manuscript.

Response: Thank you very much for your comments. We really appreciate it.